# Effect of Phenolic Compounds from Almond Skins Obtained by Water Extraction on Pork Patty Shelf Life

**DOI:** 10.3390/antiox11112175

**Published:** 2022-11-03

**Authors:** Marisa Timón, Ana Isabel Andrés, Ludovico Sorrentino, Vladimiro Cardenia, María Jesús Petrón

**Affiliations:** 1Food Technology, School of Agricultural Engineering, University of Extremadura, 06007 Badajoz, Spain; 2Department of Agricultural, Forest and Food Sciences, University of Turin, 10095 Grugliasco, TO, Italy

**Keywords:** almond by-product, aqueous extract, phenolic compound, antioxidant activity, pork patty, shelf life

## Abstract

The extraction of phenols from almond skin using water has not been applied before. The purpose of this study was to obtain aqueous extracts from almond skin to be added to pork patties to prolong their shelf life. Four different varieties of almonds were studied and aqueous extracts were obtained. The antioxidant capacity and composition of phenol compounds of the extracts were determined. Results showed that the use of water produces extracts with phenol compounds and antioxidant capacity, with the Antoñeta variety presenting the best performance in terms of antioxidant behavior. The most abundant phenolic compounds identified were isorhamentin-3-*O*-rutinoside, catechin and protocatechuic acid, all of them had a hydrophilic character due to the –OH groups in their molecules. The effect of almond skin extracts (ALMOND) on the shelf life of pork patties was compared with the effects of a control without extract (CONTROL NEG) and a control with sodium ascorbate (CONTROL POS). Throughout storage, values of pH, weight loss, headspace composition, color, TBARs and psychrotrophic aerobic bacteria were studied. CONTROL POS samples showed the lowest lipid oxidation values in comparison to CONTROL NEG or ALMOND extract samples.

## 1. Introduction

In the last 10 years, almond has been the most produced tree nut worldwide [1], and this tendency is still maintained. Almonds are produced for consumption as raw nuts and for use in the manufacturing of several products (chocolates, cookies, marzipan; prepared as almond butter, almond milk, etc.). The growing interest in almonds is based on their properties as a reliable source of nutrients, such as lipids, proteins, minerals, vitamins and dietary fiber. Moreover, almond seeds contain high concentrations of phenolic acid compounds, including benzoic and cinnamic acid derivates (vanillic, caffeic, p-coumaric and ferulic acids), flavonols (quercetin, kaempferol and isorhamnetin), anthocyanidins (delphinidin and cyanidin) and procyanidins (B2 and B3) [2].

Consequently, as a result of the current increase in the consumption of almonds, the almond industry faces the issue of high by-product stocks. This industry generates large amounts of by-products, which represent up to 80% of the unprocessed production material and consist of hulls (40–60% of total weight), shells (20–30%) and skins (4–8%) [2]. These by-products are discarded annually, representing a substantial cost for the companies and contributing to filing landfills without exploiting them. One of the easiest ways to take advantage of these residuals is by burning them to obtain energy [2]. On the other hand, other studies report innovative and alternative practices to reduce the environmental impact of these by-products through their use as soil amendments [3], in animal feeding [4,5,6], as activated carbons [7], for metal adsorption [8] or for dye adsorption [9]. In addition, some recent studies have focused on determining the total phenolic content (TPC) and antioxidant properties of the kernel and hull of almonds as another option for valorizing these by-products [10,11,12,13].

However, almond skin remains unexploited. The composition of this by-product includes total dietary fiber, soluble dietary fiber, lipids and proteins [14]. In addition, various studies have reported that it contains about 60–80% of the total phenolic compounds in the nut [2,15]. Several authors have revealed the presence of the flavonol kaempferol-3-*O*-glucoside and the flavanones isorhamnetin-3-*O*-rutinoside, isorhamnetin-3-*O*-glucoside and isorhamnetin aglycone, as well as naringenin-7-*O*-glucoside, epicatechin and catechin, as the most abundant phenolics in almond skins [16,17,18,19]. Moreover, the chemical properties of almond skin have been demonstrated, and possible alternative applications such as the recovery of phenols and their use as dietary/nutraceutical supplements have been described [17,20,21,22,23].

Most of the published research focused on the recovery of almond skin components by use of solvents such as methanol, ethanol, acetone and hexane in different conditions of concentration, pH and time of extraction [17,18,19,22,23] to define the optimal conditions for obtaining functional extracts. On the other hand, considering that the almond extracts could be directly included in food formulations, a “green” solvent as an alternative to the organic solvents could represent a better choice.

To our knowledge, no studies have focused on extracting phenol compounds from almond skin using water as a solvent. Water is the most environmentally friendly, easily available, nontoxic and inexpensive solvent [24]. In addition, it has been used as an alternative for recovering phenolic compounds from other vegetable by-products [25,26,27,28].

On the other hand, no evidence related to the possible implications of almond skin as a natural food preservation additive has been found. However, a considerable amount of by-products from the vegetable and fruit industry, rich in phenol compounds with antioxidant activity, have been utilized to inhibit lipid oxidation and microorganism growth as well as to improve color stability, especially in meat and meat products [25,28,29,30,31,32,33].

Thus, this study aimed to obtain aqueous extracts from almond skin from four different varieties for addition as a natural additive to meat patties to prolong their shelf life, after demonstrating their potential as antioxidants.

## 2. Materials and Methods

### 2.1. Chemical Standards and Reagents

The reagents sodium carbonate, di-sodium hydrogen phosphate, sodium carbonate, potassium ferricyanide, potassium persulfate, ethylenediamine tetra acetic acid (EDTA) and gallic acid were purchased from Scharlau (Barcelona, Spain). Folin–Ciocalteu’s phenol reagent, absolute ethanol and trichloroacetic acid (TCA) were purchased from Panreac (Castellar del Vallès, Barcelona, Spain). 2,2-diphenyl-1-picrylhydrazyl (DPPH•) free radical, Iron (II) Chloride 4-hydrate, 6-hydroxy-2,5,7,8-tetramethylchromane-2-carboxylacid (Trolox), (L)-Dehydroascorbic acid and malondialdehyde (MDA) were purchased from Sigma Chemical Co. (Steinheim, Germany, and St. Louis, MO, USA). Propylgallate (PG) was purchased from Acrōs Organics (Fair Lawn, NJ, USA). 2-Thiobarbituric acid (TBA) was purchased from Merck KGaA (Darmstadt, Germany). The following standards were used: phenolic acids—hydroxybenzoic acid, vanillic acid, protocatechuic acid; cinnamic acids—chlorogenic acid, 4-coumaric acid; flavanols—catechin; flavonols—kaempferol, quercetin, kaempferol-3-*O*-rutinoside, kaempferol-3-*O*-glucoside, isorhamnetin-3-*O*-rutinoside, rutin or quercetin-3-*O*-rutinoside; flavanones—naringenin, eriodictyol, eriodictyol-7-*O*-glucoside (Merck KGaA, Darmstadt, Germany).

### 2.2. Sampling

Almond fruits were harvested in August–September 2020 at the orchards of the School of Agricultural Engineering located in Badajoz (Extremadura, Spain). Four different varieties of almond (Prunus dulcis) were studied: Antoñeta, Guara, Soleta and Belona. These are four new varieties that face the issue of frost susceptibility since they blossom in February. Processing was carried out in the pilot plant of the School of Agricultural Engineering, in accordance with the usual process in an almond factory as follows: The almond hull was removed and the shell was separated from the kernel using a nutcracker. Subsequently, kernels were blanched in boiling water at 95 °C for 3 min, and later, the skin was manually removed. Finally, skins were oven-dried at 95–98 °C for 10 h until constant weight. The dried samples were packaged in vacuum bags and kept at room temperature until the extraction procedure.

### 2.3. Aqueous Extraction Procedure

Aqueous extracts from almond skin (Antoñeta (n = 5), Guara (n = 5), Soleta (n = 5) and Belona (n = 5)) were obtained following the procedure described by Andrés et al. [34]: Samples were extracted using distilled water (ratio water/skin, 20 mL:1 g). Extractions were performed using a shaking bath (J.P. SELECTA, Unitronic reciprocating shaking bath, Barcelona, Spain), at 30 °C for 60 min. Then, the samples were cooled in ice and filtered using a sieve with a 45 μm pore size (Filtra, Badalona, Spain). Afterward, the clarified liquid obtained was centrifuged at 3000× *g* rpm for 10 min at 4 °C (EPPENDORF, Centrifuge 5810-R, Madrid, Spain), and the supernatants were collected and stored at −80 °C until analysis.

### 2.4. Antioxidant Capacity

#### 2.4.1. Determination of Total Phenolic Content

Total phenolic content in the extract was estimated by spectrophotometric analyses using the Folin–Ciocalteau reagent based on procedures described by Singleton and Rossi [35], with some modifications. Briefly, 0.250 mL of sample was mixed with 0.250 mL of distilled water and 2.5 mL of 10-fold diluted Folin–Ciocalteau’s phenol reagent. Then, 2 mL of sodium carbonate 7.5% was added to the mixture and vortexed for 3 s. The reaction was kept in the dark at room temperature for 1 h, after which the absorbance was measured at 760 nm (UV-Vis biomate 3, THERMO SCIENTIFIC, Fisher Scientific, S.L., Madrid, Spain). Quantification was performed using a calibration curve prepared with gallic acid (GA), and the results were expressed as mg of gallic acid equivalents (GAEs)/g of almond skin.

#### 2.4.2. Reducing Power

The reducing power was determined according to the method of Broncano et al. [36], with some modifications. The extract (250 µL) and 250 µL of distilled water were mixed with 2.5 mL of 0.2 M phosphate buffer (pH 6.6) and 2.5 mL of 1% potassium ferricyanide (K_3_Fe(CN)_6_). That mixture was incubated at 50 °C for 20 min. Then, 2.5 mL of 10% trichloroacetic acid (*w*/*v*) was added, and samples were centrifuged at 1500× *g* rpm for 10 min (EPPENDORF, Centrifuge 5810-R, Madrid, Spain). Then, 2.5 mL of supernatant was separated and 2.5 mL of distilled water and 0.5 mL of 0.1% (*w*/*v*) ferric chloride (FeCl_3_) were added and mixed, and the absorbance was measured spectrophotometrically at 700 nm. The increase in absorbance indicates an increase in reducing power, which will be expressed in mg of Trolox equivalent antioxidant capacity (TEAC)/g of almond skin. A standard curve prepared with a Trolox solution 0.05% (*w*/*v*) was used for calculations.

#### 2.4.3. DPPH Radical-Scavenging Activity

The scavenging activity of the DPPH radical (2,2-diphenyl-1-picryl-hydracil) of the aqueous extracts was determined by the method described by Broncano et al. [36], with some modifications. A quantity of 10 µL sample and 490 µL of distilled water was added; 500 µL of absolute ethanol and 125 µL of 0.01% DPPH were added. Samples were vortexed and then kept for one hour at room temperature without light exposure. The absorbance was measured with a wavelength of 517 nm. The scavenging activity of samples was calculated using a standard curve prepared with a Trolox solution of 0.01%. Results were expressed as mg of Trolox equivalent antioxidant capacity (TEAC)/g of almond skin.

#### 2.4.4. ABTS+ Radical-Scavenging Activity

The ABTS radical elimination capacity was determined according to Swieca et al. [37], with some modifications. To obtain the ABTS+ stock solution, 7 mM ABTS solution was reacted with a 2.45 mM potassium persulfate solution, and the mixture was allowed to stand in the dark for 16 h at room temperature. The ABTS+ working solution was prepared by diluting the ABTS+ stock solution with distilled water to obtain an absorbance of 0.7 ± 0.05 at a wavelength of 734 nm. Then, 0.04 mL of the extract was added to 3 mL of ABTS+. The absorbance was measured at 734 nm after 30 min of incubation in the dark. Scavenging activity was calculated using a calibration curve prepared with a Trolox solution of 0.05% (*w*/*v*). Results were expressed as mg of Trolox equivalent antioxidant capacity (TEAC)/g of almond skin.

### 2.5. Phenol Identification and Quantification by HPLC-DAD Analysis

The extracts, previously filtered (0.45 µm), were analyzed by reversed-phase HPLC using an HP 1100 chromatograph (Agilent, Waldbronn, Germany) coupled with a diode array detector (Agilent, Waldbronn, Germany) and an Inertsil ODS-3 column (5.0 μm particle size, 4.6 mm inner diameter × 250 mm) preceded by an Inertsil ODS-3 pre-column (5.0 μm, 4.0 mm × 10 mm). A mobile phase consisting of 2.5% formic acid in water (phase A) and 2.5% formic acid in acetonitrile (phase B) with a flow rate of 1 mL/min was used under the following gradient: 0 min, 5% B; 13 min, 11% B; 16 min, 13% B; 20 min, 14% B; 22 min, 15% B; 25 min, 20% B; 28 min, 25% B; 30 min, 30% B; 40 min, 5% B. Runtime was 40 min. The injection volume was 25 µL, and analytes were detected at a wavelength of 280 nm.

The identification of phenolic compounds was performed by comparing the retention times with those of standards, and the quantification was carried out with calibration curves.

### 2.6. Effect of Aqueous extracts of Almond Skin on Shelf Life of Fresh Pork Patties

#### 2.6.1. Pork Patties Preparation and Packaging

Ground meat was provided by a local butchery, and pork patties were made and weighed to obtain samples of 80 ± 1 g each. To study the effect of almond skin extracts on pork patty shelf life, three experimental batches were prepared: (1) control batch without extract (20 mL of distilled water/kg meat) (CONTROL NEG) (n = 5); (2) positive control batch, with sodium ascorbate added as antioxidant (20mL sodium ascorbate (20 mL:1 g)/kg meat) (CONTROL POS) (n = 5); (3) extract batch, with extract obtained from Antoñeta almond skin (20 mL of extract/kg meat) (ALMOND) (n = 5). Synthetic antioxidant and extract were added to the meat in quantities of 1 g/1 kg meat. Pork patties were packaged in polypropylene trays (130 × 160 × 50 mm^3^, Sarabia Plastics, Alicante) and sealed with a polyester methyl cellulose PLPMC film (Wipack, Hamburg, Germany) with oxygen permeability of 114 cm^3^/cm^2^/24 h. Packaging equipment was used (Smart 500, ULMA, Sevilla, Spain) to introduce a mixture of O_2_ and CO_2_ gases of 64.7% and 28.6%, respectively. The trays were stored at 4 °C. Samples were analyzed immediately and after 3, 7 and 9 days of storage.

#### 2.6.2. pH Analysis

The pH determination was carried out using a puncture pH-meter specific for meat products (HANNA, model HI 99163, Instrumentación Científica y Técnica, S.L, La Rioja, Spain). The pH was measured in fresh meat on days 0, 3, 7 and 9 of packaging and storage in refrigeration.

#### 2.6.3. Weight Loss Analysis

Pork patties were weighed on a technical balance before and after days 3, 7 and 9 of refrigerated storage in order to calculate the weight loss of the meat. Weight loss was calculated using the following formula:Weight loss (%) = ((W_0_ − W_analysis day_)/W_0_) × 100(1)

W_0_ is the weight of meat on day 0, and W_analysis day_ is the weight of meat after 3, 7 or 9 days.

#### 2.6.4. Analysis of the Composition of Gases in the Headspace

The gas composition of the headspace of trays during storage was analyzed by means of a gas analyzer (Checkpoint, PBI Dansensor, Ametek Instrumentos, SLU, Barcelona, Spain). Each sample was punctured on days 0, 3, 7 and 9, and the percentages were recorded.

#### 2.6.5. Instrumental Color Evaluation

The color measurement was carried out on the surface of the fresh meat burgers after the tray opening on days 0, 3, 7 and 9. Two measurements were made in different areas of the meat surface. The coordinates of brightness (L*), the intensity of the red color (a*, red + -green) and intensity of the yellow color (b*, yellow + -blue) were determined using a colorimeter (Minolta Mod. Chroma Meter CR-400, Aquateknica, SA, Valencia, Spain).

#### 2.6.6. Measurement of Lipid Oxidation by TBARS

The analysis was conducted according to the procedure described by Jørgensen and Sørensen [38], with some modifications. Samples were ground, and 5 g was weighted and transferred to a centrifuge tube where 15 mL of 7.5% trichloroacetic acid (TCA) solution, containing 0.1% propylgallate (PG) and 0.1% ethylenediaminetetraacetic acid disodium salt (EDTA), was added and mixed with an Ultra-Turrax mixer (IKA-Werke GmbH & Co. KG, Staufen, Germany) for 30 s at 5000 rpm. Afterward, the whole solution was centrifuged for 15 min at 12,000× *g* rpm, to separate the liquid from the solid phase. 3 milliliters was withdrawn to a test tube and 3 mL of 0.02 M TBA reagent was added and vortexed for 3 s. The test tube with the sample was heated at 92–96 °C in a warm water bath for 40 min. Subsequently, samples were cooled down in cold water (made with a water bath with ice) until reaching the temperature of 4 °C. Then, as some samples were blurred, they were centrifuged for 10 min at 4500× *g* rpm in glass tubes at 6 °C. If bubbles appear in the test tubes, they may be removed by ultrasonic treatment. The absorbance was measured at wavelengths of 532 nm, the maximum absorbance of the TBA-MDA complex, and 600 nm against the correction for nonspecific turbidity. The content of TBARS was calculated using a standard curve prepared with a 1,1,3,3-tetraethoxypropane (TEP). Results were expressed as mg malondialdehyde (MDA)/kg sample.

#### 2.6.7. Psychrotrophic Microbial Analysis

Twenty-five grams of each sample was taken aseptically and homogenized with 90 mL of peptone water in a laboratory blender (Stomacher 400 Circulator, Instrumentación Científica y Técnica, S.L, La Rioja, Spain). Serial decimal dilutions were performed in sterile peptone water, and 1 mL samples of appropriate dilutions were poured or spread onto a standard Plate Count Agar (PCA) and then incubated at 7 °C for 10 days. Results were expressed as log_10_ CFU (colony forming units)/g.

Finally, *Salmonella* spp. and *L. monocytogenes* were determined according to ISO 6579 [39] and ISO 11290-1 [40], respectively, using Xylose Lysine Deoxycholate (XLD) and Modified Oxford (MOX) agar, at 35 °C for 48 h.

### 2.7. Statistical Analysis

Analysis of variance was used to test any difference in antioxidant activities and phenolic profile among skin extracts of almond varieties and to study the effect of Antoñeta almond extract on pork patty shelf life. Tukey’s test at 95% confidence level was used to determine different means between treatments. Correlations were calculated using Pearson’s correlation coefficient (r). All these analyses were performed using the SPSS v.21.0 (IBM-SPSS, Chicago, IL, USA) software.

## 3. Results and Discussion

### 3.1. Antioxidant Capacity

#### 3.1.1. Total Phenolic Content

As reported in Table 1, values of phenol contents detected in samples were generally lower than those reported in the literature, where organic solvents were used. The Antoñeta variety presented the highest value (2.25 mg GAE/g), followed by Belona (2.04 mg GAE/g) and Soleta (1.77 mg GAE/g). In Guara, a content of 1.32 mg GAE/g was found. However, similar values were obtained by Smeriglio et al. [17] in almond skin using methanol and ethyl acetate as solvents. On the contrary, Progmet et al. [18] reported values between 7.62 and 25.17 mg GAE/g dw (dry weight) in almond skin using methanol/water (70:30, *v*/*v*); Valdés et al. [16] found values around 100 mg QE (quercetin)/g skin using ethanol in Guara almonds. On the other hand, Monagas et al. [21], who carried out the extraction with acetone, reported a content of phenols of 22.8 mg GAE/g of skin, and Barreira et al. [41] obtained values between 163.71 and 9.22 mg GAE/g using methanol as a solvent. In view of these results, a higher effectiveness of organic solvents as compared to water is evident for extracting phenols, and the use of environmentally friendly, nontoxic and cheap solvents is needed nowadays [18]. In this sense, the use of water as a solvent in combination with other techniques such as the use of natural deep eutectic solvents (NADESs), maceration, ultrasound-assisted extraction and homogenate-assisted extraction was very successful for the recovery of phenolic compounds, and these methods exhibited a higher efficiency for the extraction of these compounds in comparison to conventional solvents such as methanol and ethanol [42].

Regarding almond varieties, significant differences (*p* ≤ 0.05) were found (Table 1), with Antoñeta almond skin showing the highest values of phenols and Soleta the lowest. This cultivar shows one of the earliest ripenings in comparison to the others. In this sense, Bolling et al. [43] have reported the effect of cultivar and the agro-climatic conditions on the concentration of polyphenols in almond skins, both factors having a significant impact. The same conclusions were achieved by Barreira et al. [41] in a study with ten regional and commercial Portuguese almond cultivars.

#### 3.1.2. Reducing Power, DPPH and ABTS Radical-Scavenging Activities

Results of reducing power (RP) ranged between 4.15 and 7.80 mg TEAC/g sample. DPPH analysis reached values between 0.82 and 1.81 mg TEAC/g sample, and the ABTS method registered values from 3.85 to 6.24 mg TEAC/g sample (Table 1). In all cases, Antoñeta extracts showed the highest values of these parameters (*p* ≤ 0.05), also confirming the effect of cultivar and agro-climatic conditions on the antioxidant activity of almonds [41,43,44]. Other authors also found antioxidant activity in almond skin [16,17,18,21]. Nevertheless, comparison among results from different studies is not always possible since tests and standards used to express the antioxidant activity are unalike. Progmet et al. [18,19] described lower values of RP, DPPH and ABTS activities in almond skin extracts compared to values in the present study. In the same sense, values of ABTS activity were also lower in the study of Smeriglio et al. [17] than in the current experiment. In view of the results of those authors, it seems that a higher content of phenols in almond skins does not necessarily imply a higher antioxidant activity. In this sense, Petrón et al. [45] suggested that the Folin–Ciocalteu assay to measure phenol content not only reflects phenol amount, since the F–C reagent could also react with other compounds such as proteins, carbohydrates, amino acids, unsaturated fatty acids, vitamins, amines, aldehydes and ketones, thus possibly giving an overestimation of phenols [46,47]. On the contrary, other authors showed higher values of RP together with higher phenol content in almond skin cultivars [16].

In the present work, positive and statistically significant correlations between total phenol content and antioxidant activity measured by RP, DPPH and ABTS methods were found (r = 0.843 *p* < 0.01, r = 0.891 *p* < 0.01 and r = 0.885 *p* < 0.01, respectively). Other studies have also reported a strong correlation between total phenolic compounds and antioxidant capacity in almond skin [17,41]. Therefore, it could be suggested that total phenol content could be a good indicator of the antioxidant activity of almond skin extracts.

### 3.2. Phenol Profile of Extracts—Identification and Quantification by HPLC-DAD Analysis

The antioxidant activity of the extracts should be related not only to the total phenol content but also to the individual polyphenols present in these extracts, as was suggested by Smeriglio et al. [17]. HPLC analysis was carried out to identify these compounds. Compounds observed in the present study in almond skin were also identified by other authors [16,17,18]. The most abundant phenolic compounds were isorhamentin-3-*O*-rutinoside, catechin and protocatechuic acid (Table 2). It can be observed that isorhamnentin-3-*O*-rutinoside is the phenolic compound present in the highest quantities in all extracts (ranging from 75.82 to 134.08 µg/g sample). Similar results were reported in almond skins of different cultivars in other studies [16,48,49]. These authors suggested that the presence of a high number of –OH functional groups in the molecule of the isorhamnentin-3-*O*-rutinoside led to an increased hydrophilic character and consequently a high solubility in water. Therefore, it is expected that the use of water as a solvent in the current study promotes a high extraction level of this compound. Isorhamnentin-3-*O*-rutinoside is derived from the quercetin molecule by methylation [50]. Quercetin has a broad range of biological activities, including antioxidant, analgesic, anti-inflammatory, cardio- and neuroprotective and antiallergic activities [51].

Catechin content ranged from 36.32 to 47.85 µg/g sample. Pronadov et al. and Bartolomé et al. [52,53] also found the highest quantities of this compound in almond skin extracts using acetone/water (50/50, *v*/*v*) and methanol/HCl (1000/1, *v*/*v*) as solvents, respectively. Catechin also presents a hydrophilic character due to the five –OH groups in its molecule [16], which would explain its high recovery using water or other polar solvents. However, other authors did not find this compound in almond skin or found it just at trace levels despite the use of methanol or ethanol as extracting solvents [17,18,19].

Values of protocatechuic acid, a dihydroxybenzoic acid, were also high, ranging between 20.87 and 45.01 µg/g sample. Progmet et al. [18] also reported high content of this compound in blanching water used when almond skin was removed, and this fact shows the affinity of this benzoic acid for dissolution in water. However, when organic solvents were used for the extraction, lower quantities of dihydroxybenzoic acid were found [17,53].

On the other hand, compounds such as naringenin or 4-coumaric acid were not identified in the present study (4-coumaric only appeared in Guara variety). These compounds were found in high quantities in almond skin extracts obtained using organic solvents [17,19,49]. In this sense, Valdés et al. [16] described a hydrophobic character and low solubility in water of the naringenin compound due to the low amount of –OH groups in the molecule; this explanation justifies the absence of this compound in this study. However, the considerable presence of compounds such as kaempferol-3-*O*-glucoside, kaempferol-3-*O*-rutinoside, 4-hydroxybenzoic acid, quercetin-3-*O*-rutinoside and eriodictyol-7-*O*-glucoside, which were also identified by other authors with similar quantities in extracts obtained using organic solvents [16,17,19,52,53], in almond skin extracts in this study reveals the effectiveness of water as a solvent for the general extraction of phenols in almond by-products.

Regarding the effect of the almond variety on phenol profile, Guara almond extracts showed the highest isorhamnentin-3-*O*-rutinoside, kaempferol-3-*O*-glucoside and eriodictyol-7-*O*-glucoside contents compared to the rest of varieties. On the other hand, the Antoñeta variety displayed the greatest amount of protocatechuic acid, catechin and quercetin-3-*O*-rutinoside. Valdés et al. [16] suggested that genetic variation was likely responsible for the different phenol profiles among cultivars. In view of these results, it is difficult to relate the phenol profiles of different extracts to the antioxidant activity of the samples. Thus, a more intense antioxidant activity in Antoñeta samples (Table 1) could not be entirely ascribed to a specific phenol compound or several phenol compounds. Further studies should be carried out to elucidate that topic.

### 3.3. Effect of Aqueous Extracts of Almond Skin on Shelf Life of Fresh Pork Patties

Based on the antioxidant results of this study (Table 1), the aqueous extract from Antoñeta almond skin was selected to be tested as an additive in pork patties in order to study the effect on the shelf life of this meat product. A positive control based on sodium ascorbate was also used for comparison.

#### 3.3.1. Headspace Composition, pH and Weight Losses

Table 3 shows the headspace composition of packs containing pork patties as well as pH and weight loss during storage. The initial O_2_ content was 67.93 ± 0.34%, and the O_2_ content significantly decreased (*p* ≤ 0.05) for all treatments during 9 days under storage conditions. The final O_2_ values ranged from 58.75% to 63.50%. The depletion of oxygen content in the headspace of modified atmosphere containers of meat products has been previously described by other researchers [54,55], who related the decrease in O_2_ to the participation of this gas in oxidation reactions or microbial growth. The CO_2_ concentration, initially set at 25.60%, showed an increasing trend during storage, reaching final values ranging between 29.10% and 30.55% (*p* < 0.05). Regarding the treatment effect, in general, neither O_2_ nor CO_2_ content appeared to show significant differences among samples (*p* > 0.05). In this sense, Andrés et al. [25] also found no differences in O_2_ and CO_2_ contents of lamb patties with different by-product extracts during storage using a modified atmosphere.

pH values (Table 3) were consistent with those provided by other authors for raw pork patties packed in MAP [29,56]. No significant differences were observed for this parameter during storage (*p* > 0.05). This is remarkable since an increase in pH could be ascribed to an increase in microbial counts [29], which in fact was reported in the present research (Section 3.3.4). On the other hand, CONTROL NEG samples showed the highest pH values throughout storage (*p* ≤ 0.05). It could be hypothesized that the relatively low pH values of almond extracts (due to their composition) and sodium ascorbate could have affected the pH of ALMOND and CONTROL POS samples, reducing the pH in comparison to that of CONTROL NEG samples.

Values of weight loss (%) ranged from 0.32% to 1.19%, and these values are considered very low [25]. Weight loss of meat during storage is related to the quality of products since it affects sensory parameters such as color, tenderness, juiciness and overall appearance; hence, large losses would cause a loss of quality in meat products [25]. Even though weight losses were low, they significantly increased during the storage of samples from day 0 to day 9 (*p* ≤ 0.05) in all treatments. CONTROL NEG samples showed the highest increase in weight loss, reaching the highest values after 9 days (1.19%) (*p* ≤ 0.05). It could be suggested that the use of almond extracts in pork patties reduces weight losses in samples, which could be related to an improvement in water-holding capacity in samples with added antioxidant extracts. In this sense, Yueyue et al. [57] described a decrease in water-holding capacity associated with higher oxidation, specifically in proteins, in bighead carp fillets. However, other authors could not find this effect on weight loss in meat patties using different antioxidant vegetable extracts [25,29,58].

#### 3.3.2. Instrumental Measurement of Color

Values for color parameters and the effect of storage and extract addition on pork patties are shown in Table 4.

Changes in parameters L* and a* were significant throughout storage (*p* < 0.05), with L* values showing an opposite trend to a* values. L* followed an increasing trend in all treatments, from initial values in the range 39.52–40.52 to final values in the range 43.90–49.00 (*p* ≤ 0.05). The increase in lightness during storage has been related to oxidation reactions and the availability of water on the surface of the patties, producing a greater dispersion of light, which has been reported by several authors [59]. On the contrary, other studies showed constant L* values during the storage of pork patties with added antioxidant extracts [29,30]. Regarding the effect of the different treatments on L*, this was only statistically significant after 9 days of storage, with patties with almond skin extracts showing the highest values (*p* < 0.05).

Concerning a* values, these decreased throughout storage in all the treatments (*p* ≤ 0.05), from initial values in the range 18.88–19.80 to final values in the range 9.45–12.57. The decline in redness of meat is mainly due to the oxidation of myoglobin forming metmyoglobin, which is brown, and it has been described by a considerable number of researchers [25,30,59]. No significant effect of treatment was observed for a* values during storage (*p* > 0.05).

On the other hand, values of b* only decreased throughout storage in ALMOND samples (*p* < 0.05). Tamkuté et al. [30] also described that the loss of meat redness and transition to brownish red color resulted in a decrease in b* values.

As shown in Table 4, the addition of almond extract did not significantly influence the color of samples in this study (*p* > 0.05), hence making the use of this extract very promising as an alternative additive, since no variation for L*, a* or b* was produced. There is no scientific research to compare these results with, though other authors described a significant effect of other vegetable extracts on the color of fresh patties, modifying their appearance [29,30,59].

#### 3.3.3. Measurement of Lipid Oxidation by Thiobarbituric Acid Reactive Substances (TBARs)

TBAR values, expressed as mg MDA/kg meat, significantly increased from initial values of 0.06–0.12 to final values of 0.15–0.52 (*p* ≤ 0.05) (Table 4), representing the progression of lipid oxidation. Previous studies have also described a significant increase in MDA values during chilled storage of pork patties [25,29,33]. CONTROL POS samples showed the lowest oxidation in comparison to ALMOND and CONTROL NEG samples (*p* ≤ 0.05) throughout this study; therefore, sodium ascorbate exhibited a higher antioxidant effect than almond extracts. Other authors found an opposite trend in pork patties with synthetic antioxidants (sodium ascorbate or BHT), in comparison to extract-added samples [29,59]. These studies showed that extracts of oak wood or guarana seed possessed a higher antioxidant effect compared to synthetic ones. Moreover, Andrés et al. [25] also found lower values of TBARs in lamb patties when grape or olive pomace extracts were added, and extracts were more effective in samples than sodium ascorbate. As a potential antioxidant effect was demonstrated in almond extracts, it could be thought that the addition of a greater concentration in samples could enhance the lipid oxidation inhibition. In this way, Šojić et al. [32] demonstrated that different doses of wild thyme extracts implied differences in the lipid oxidation of pork patties. However, the last hypothesis could fail since the presence of MDA in almond extracts was confirmed (0.000043 mg MDA/mL) before the extracts were used in pork patties; this fact probably reduced the oxidative stability of meat samples.

#### 3.3.4. Psychrotrophic Microbial Analysis

Results showed the absence of pathogens (*L. monocytogenes* and *Salmonella* spp. in 25 g of meat) at the beginning of storage, so they were not again analyzed during the following days of storage.

Changes in psychrotrophic aerobic bacteria during the storage of pork patties are depicted in Figure 1. Initial counts were 2.78, 2.65 and 2.20 log10 cfu/g in ALMOND, CONTROL POS and CONTROL NEG, respectively, all of them lower than microbiological limits established by the International Commission for Microbial Specifications in Food (ICMSF) in fresh meat [60]. A significant increase in counts (*p* ≤ 0.05) was observed for all treatments throughout storage (*p* < 0.05). At the end of the storage, counts ranged from 3.35 to 3.38 log_10_ cfu/g.

Regarding the effect of treatments on microbial counts, this was not statistically significant during storage (*p* > 0.05), with the only exception of samples on day 0. A possible explanation for this difference among samples may be attributed to the almond extraction process that might have contributed to the psychrotrophic microorganism count. Similarly, Pateiro et al. [59] found no effects of guarana seed extracts on total microorganisms when they studied pork patties, and Zamuz et al. [61] found no effects of hull, bur and leaf chestnut extracts on antimicrobial activity in beef patties. In another study, Sadeghinejad et al. [31] suggested that a higher concentration of pistachio green hull extract would be required for an inhibitory effect on microorganisms in beef patties during chilled storage, which could also be the case in the present research.

## 4. Conclusions

In view of the results, the effectiveness of water as a solvent to extract phenolic compounds from almond skin can be stated, and contents of phenolic compounds close to 2 mg GAE/g sample were obtained. The feasibility of using an economical and environmentally friendly procedure to extract compounds from an underexploited material to be used as natural food preservation additives, dietary and nutraceutical supplements and active ingredients for cosmetics is very relevant, representing a competitive advantage over nonaqueous extracts. The procedure could be implemented on an industrial scale and would represent an alternative for the valorization of these almond residues, which is of great interest for reducing their environmental impact, while enhancing the competitiveness and sustainability of the process. Further research would be needed to scale up this experiment to the almond industry.

Clearly, the effectiveness of water in the extraction of phenolic compounds could be improved by combining it with other techniques, as was described in other studies. Therefore, it is urgent to explore possible techniques to be combined with this green solvent for a highly efficient extraction of phenolic compounds.

The phenolic compound content and phenolic profile, as well as the antioxidant activity of aqueous extracts, were significantly affected by the almond variety, showing a significant impact of the cultivars on these parameters. Extracts of Antoñeta almonds showed the highest phenolic compound content and antioxidant activity values. This finding could be considered in the choice of cultivars to be grown in Extremadura.

The phenolic compound content of the extracts has been positively correlated to their antioxidant activity (RP, DPPH and ABTS) (r = 0.843 *p* < 0.01, r = 0.891 *p* < 0.01 and r = 0.885 *p* < 0.01, respectively), even though it was difficult to relate this activity to the phenol profile. It would be interesting to correlate the phenolic compound content to the antimicrobial activity. Further research should be undertaken to elucidate these aspects of almond extracts.

The effectiveness of almond skin extracts as an additive in pork patties has been shown to be limited. Only pH, weight loss and L* values were affected by the use of almond extracts. In this sense, new experiments should be designed to optimize the use of these extracts in meat products or other food systems, to implement the use of this almond extract as a natural additive in the food industry.

## Figures and Tables

**Figure 1 antioxidants-11-02175-f001:**
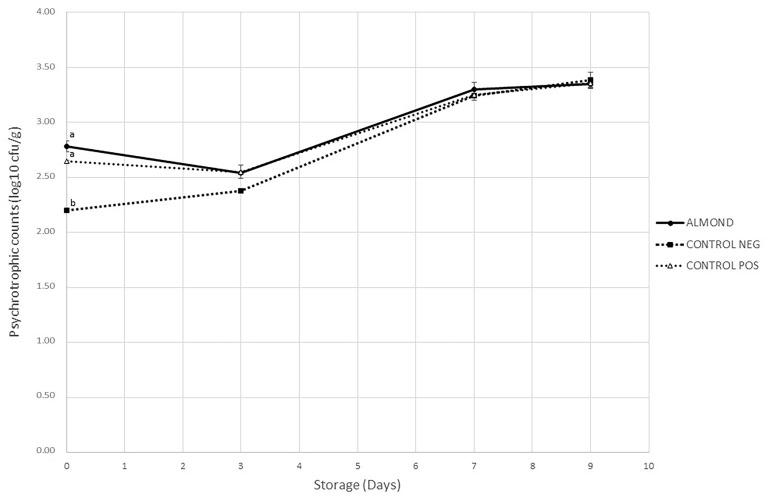
Evolution of psychrotrophic aerobic bacteria counts (log_10_ cfu/g sample) in pork patties with different treatments stored in MAP for 9 days. Values are reported as mean ± SD (n = 5). a, b values with different letters on the same day of storage are significantly different (*p* < 0.05), Tukey test.

**Table 1 antioxidants-11-02175-t001:** Total phenolic content (TPC), expressed as mg GAE/g sample, and antioxidant activity as determined by reducing power (RP), DPPH radical-scavenging activity (DPPH) and ABTS radical-scavenging activity (ABTS) methods, expressed as mg TEAC/g sample. Values are reported as mean ± standard deviation (SD) of independent replicates (n = 5).

Almond Variety	TPC	RP	DPPH	ABTS
Antoñeta	2.25 ± 0.22 ^a^	7.80 ± 1.10 ^a^	1.81 ± 0.30 ^a^	6.24 ± 0.87 ^a^
Guara	1.32 ± 0.08 ^b^	4.15 ± 0.38 ^b^	0.82 ± 0.07 ^b^	3.85 ± 0.08 ^b^
Soleta	1.77 ± 0.15 ^ab^	4.99 ± 0.51 ^ab^	1.12 ± 0.19 ^ab^	4.71 ± 0.47 ^ab^
Belona	2.04 ± 0.26 ^ab^	5.04 ± 0.68 ^ab^	1.26 ± 0.15 ^ab^	4.83 ± 0.38 ^ab^
P_varieties_	*	*	*	*

^a^, ^b^ Values with different letters are significantly different (*p* < 0.05), Tukey test. Significant levels: * *p* < 0.05 (significant difference).

**Table 2 antioxidants-11-02175-t002:** Identification and content of phenolic compounds (µg/g of sample) in the aqueous extract of Antoñeta, Belona, Guara and Soleta almonds by HPLC-DAD. Values are reported as mean ± SD (n = 5).

	Antoñeta	Belona	Guara	Soleta	*p*
Protocatechuic acid	45.01 ± 19.94	23.17 ± 7.45	20.87 ± 7.64	22.29 ± 11.51	n.s.
Hydroxybenzoic acid	11.59 ± 8.56	13.20 ± 3.98	20.89 ± 5.96	11.06 ± 9.30	n.s.
Catechin	47.85 ± 11.28	36.32 ± 0.85	46.96 ± 13.38	37.47 ± 12.33	n.s.
4-Cumaric acid	n.d.	n.d.	2.49 ± 0.75	n.d.	*
Eriodictyol-7-*O*-glucoside	9.04 ± 1.06 ^b^	5.69 ± 1.93 ^c^	12.26 ± 0.95 ^a^	6.93 ± 1.29 ^bc^	*
Quercetin-3-*O*-rutinoside	14.72 ± 5.58 ^a^	8.72 ± 1.66 ^ab^	10.46 ± 4.08 ^ab^	5.35 ± 0.56 ^b^	*
Kaempferol-3-*O*-rutinoside	13.54 ± 1.92 ^ab^	9.85 ± 2.99 ^bc^	6.35 ± 0.39 ^c^	16.72 ± 4.18 ^a^	*
Isorhamnentin-3-*O*-rutinoside	75.82 ± 2.39 ^b^	77.07 ± 2.83 ^b^	134.08 ± 58.07 ^a^	79.55 ± 1.99 ^b^	*
Kaempferol-3-*O*-glucoside	19.99 ± 1.05 ^ab^	13.58 ± 1.41 ^b^	23.02 ± 5.40 ^a^	17.25 ± 5.14 ^ab^	*

^a^, ^b^, ^c^ Values with different letters are significantly different (*p* < 0.05), Tukey test. Significant levels: n.s. (not significant); * *p* < 0.05 (significant difference). n.d. (no detected).

**Table 3 antioxidants-11-02175-t003:** Evolution of oxygen (%) and carbon dioxide (%) content in headspace, pH and weight loss (%) of packs of pork patties of different treatments stored in MAP for 9 days. Values are reported as mean ± SD (n = 5).

Day	Treatment	pH	Weight Loss	O_2_	CO_2_
0	ALMOND	5.82 ± 0.01 ^b^		67.93 ± 0.34 ^1^	25.60 ± 0.23 ^3^
	CONTROL NEG	6.02 ± 0.05 ^a^		67.93 ± 0.34 ^1^	25.60 ± 0.23 ^4^
	CONTROL POS	5.82 ± 0.02 ^b^		67.93 ± 0.34 ^1^	25.60 ± 0.23 ^2^
	P_treatment_	*		n.s.	n.s.
3	ALMOND	5.89 ± 0.01 ^b^	0.32 ± 0.04 ^b 3^	67.37 ± 0.67 ^1^	26.57 ± 0.26 ^b 2^
	CONTROL NEG	6.00 ± 0.02 ^a^	0.47 ± 0.01 ^ab 3^	67.23 ± 0.47 ^1,2^	27.00 ± 0.12 ^ab 3^
	CONTROL POS	5.94 ± 0.01 ^b^	0.48 ± 0.06 ^a 3^	67.99 ± 0.33 ^1^	27.57 ± 0.22 ^a 1,2^
	P_treatment_	*	*	n.s.	*
7	ALMOND	5.81 ± 0.00 ^b^	0.48 ± 0.03 ^b 2^	67.53 ± 0.37 ^1^	28.77 ± 0.09 ^1^
	CONTROL NEG	5.95 ± 0.02 ^a^	0.81 ± 0.10 ^a 2^	66.17 ± 0.09 ^2^	29.23 ± 0.32 ^2^
	CONTROL POS	5.93 ± 0.03 ^b^	0.60 ± 0.07 ^a 2^	64.00 ± 2.02 ^1^	28.20 ± 0.85 ^1^
	P_treatment_	*	*	n.s.	n.s.
9	ALMOND	5.82 ± 0.01 ^b^	0.62 ± 0.02 ^b 1^	63.50 ± 1.30 ^a 2^	29.10 ± 0.40 ^1^
	CONTROL NEG	6.01 ± 0.02 ^a^	1.19 ± 0.02 ^a 1^	63.15 ± 0.45 ^a 3^	30.55 ± 0.15 ^1^
	CONTROL POS	5.95 ± 0.06 ^b^	0.71 ± 0.02 ^b 1^	58.75 ± 0.15 ^b 2^	29.25 ± 0.65 ^1^
	P_treatment_	*	*	*	n.s.

^a^, ^b^ Values with different letters on the same day of storage are significantly different (*p* < 0.05), Tukey test. ^1^, ^2^, ^3^, ^4^ Values with different numbers for the same treatment are significantly different (*p* < 0.05) compared to the different shelf life days, Tukey test. Significant levels: n.s. not significant; * *p* < 0.05 (significant difference).

**Table 4 antioxidants-11-02175-t004:** Evolution of instrumental color parameters (L*, a* and b*) and TBARS (expressed as mg MDA/kg sample) in pork patties with different treatments stored in MAP for 9 days. Values are reported as mean ± SD (n = 5).

Day	Treatment	L*	a*	b*	TBARS
0	ALMOND	39.55 ± 2.34 ^3^	19.80 ± 0.89 ^1^	24.15 ± 0.67 ^1^	0.12 ± 0.02 ^3^
	CONTROL NEG	39.52 ± 0.86 ^2^	19.79 ± 0.48 ^1^	22.82 ± 0.06	0.10 ± 0.02 ^3^
	CONTROL POS	40.52 ± 0.79 ^3^	18.88 ± 0.78 ^1^	21.21 ± 0.40	0.06 ± 0.01 ^2^
	P_treatment_	n.s.	n.s.	n.s.	n.s.
3	ALMOND	40.94 ± 1.75 ^3^	18.25 ± 1.44 ^1^	23.12 ± 0.92 ^1,2^	0.18 ± 0.01 ^b 3^
	CONTROL NEG	44.94 ± 0.90 ^1^	17.05 ± 0.69 ^1,2^	21.66 ± 1.17	0.22 ± 0.01 ^a 2^
	CONTROL POS	43.45 ± 0.71 ^2^	17.05 ± 0.61 ^2^	20.72 ± 0.52	0.09 ± 0.01 ^c 2^
	P_treatment_	n.s.	n.s.	n.s.	*
7	ALMOND	45.82 ± 1.19 ^2^	13.72 ± 0.85 ^2^	22.99 ± 0.77 ^1,2^	0.31 ± 0.06 ^ab 1,2^
	CONTROL NEG	46.04 ± 3.13 ^1^	12.13 ± 2.06 ^2^	20.80 ± 1.27	0.32 ± 0.12 ^a 1,2^
	CONTROL POS	44.41 ± 0.65 ^1^	14.38 ± 0.40 ^3^	21.71 ± 0.16	0.19 ± 0.11 ^b 1^
	P_treatment_	n.s.	n.s.	n.s.	*
9	ALMOND	49.00 ± 0.08 ^a 1^	9.45 ± 0.43 ^3^	20.33 ± 0.14 ^2^	0.51 ± 0.08 ^ab 1^
	CONTROL NEG	44.34 ± 0.66 ^b 1^	11.89 ± 1.39 ^3^	20.26 ± 1.75	0.52 ± 0.07 ^a 1,2^
	CONTROL POS	43.90 ± 0.75 ^b 1^	12.57 ± 0.57 ^4^	21.31 ± 0.88	0.15 ± 0.02 ^b 1^
	P_treatment_	*	n.s.	n.s.	*

^a^, ^b^, ^c^ Values with different letters on the same day of storage are significantly different (*p* < 0.05), Tukey test. ^1^, ^2^, ^3^, ^4^ Values with different numbers for the same treatment are significantly different (*p* < 0.05) compared to the different shelf life days, Tukey test. Significant levels: n.s. not significant; * *p* < 0.05 (significant difference).

## Data Availability

The data are contained within the article.

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
