# Peer review of "Effect of Phenolic Compounds from Almond Skins Obtained by Water Extraction on Pork Patty Shelf Life"

_antioxidants, 2022, doi:10.3390/antiox11112175_

Round 1
Reviewer 1 Report (Previous Reviewer 3)
The authors made most of my suggestions and reply to the rest by acceptable responce
Author Response
Please see the attachment

Reviewer 2 Report (New Reviewer)
The use of almond skin water extract as a preservative in meat by-products is an interesting topic.
In the manuscript, the introduction gave the necessary background and the methods are rigorose descripted. The results are clearly described. The corrections made were essential to complete the work
Author Response
Please see the attachment

Reviewer 3 Report (New Reviewer)
(1) Title, replace "green" by water.
(2) In the part of 2.5, what are commercial standards?
(3) In the part of 3.1.1, as for the text "Similar results were obtained by Semeriglio et al.", what similar results should be given. Otherwise, the text is not consistent with the text "values of phenol contents detected in samples were generally lower than those reported in literature, where organic solvents were used".
(4) Conclusions, the main results can be added.
Author Response
Please see the attachment

Reviewer 4 Report (New Reviewer)
Authors describe the effecet of of phenolic compounds from almond skins obtained by green extraction on pork patties shelf life.
The experimental design is correct as well as the approach however I believe that to complete the study sensorial analysis is necessary in order to evaluate the impact of the addition on the quality of meat and consumer acceptance.
For this reason I believe that authors should added these data to complete their work.
Moreover some corrections are necessary in the text:
1. The abstract should be rewritten in a more concise way and less descriptive
2. insert picture of pork patties during shelf life
3. Which positive control was used in antioxidant activity?
4. Rewrite conclusion section taking into account these observations:
What is the advantage of using almond waste rather than other extracts with antioxidant activity? How many days has the shelf life been extended compared to other systems?
Round 2
Reviewer 4 Report (New Reviewer)
Authors correctly reply to some queries however the did not insert sensorial analysis.
Unfortunatly I believe that to complete the study sensorial analysis is necessary in order to evaluate the impact of the addition on the quality of meat and consumer acceptance. For this reason I have asked to the authors to added these part to complete their work. For this reason manuscript was not suitable for publication in this form.
Author Response
Please see the attachment

This manuscript is a resubmission of an earlier submission. The following is a list of the peer review reports and author responses from that submission.
Round 1
Reviewer 1 Report
This work uses water as the extraction solvent to obtain phenolic compounds from almond skin. Skin from four varieties of almond, Antoñeta (n=5), Guara (n=5), Soleta (n=5), and Belona (n=5), was compared in the total phenol content (TPC), Fe3+ reducing power, DPPH radical-scavenging activity, ABTS+ radical-scavenging activity, and phenol distribution (analyzed by HPLC-DAD). The Antoñeta variety was identified to contain the highest level of TPC and thus was used as a preservative in fresh pork patties compared with sodium ascorbate. The quality of the pork patties was assessed over 0, 3, 7, and 9 days period using the parameters of pH, weight loss analysis, O2 and CO2 in the headspace, color measurement, lipid oxidation (by TBARS), and psychrotrophic microbial growth (colony count, long10 CFU). Although the subject of research had some merit, there were serious flaws in the experimental design and a lack of clarity in the presentation. Additionally, numerous errors in the use of standard English style and typos made the manuscript difficult to follow.
- Introduction: more background information is needed on the known differences in the TPC and antioxidative properties among the various almond varieties, especially if there are any publications on the specific varieties used in this work. It remains unclear why these varieties were chosen for the study.
- Experimental design: clearly, the work was intended to test using water as the alternative solvent to extract phenolic compounds in almond skin. By doing so, naturally, only water-soluble compounds would be extracted. In the absence of proper control, such as those fairly complete extractions of phenolic compounds, it's difficult to assess the extraction efficiency by water and the properties of the extracted compounds, including those examined in this work. Therefore, it remains unknown what the advantages are of using water for the extraction compared to other extraction methods, other than the obvious reasons (environmental), as stated.
- The effect of heat on the phenol content should be addressed. Previous work such as that by Guiné et al., Int. J. Fruit Sci., 2015, 15, 173-86 and Xu et al., J. Agric. Food Chem., 2007, 55, 330-5 has already suggested that heat can significantly affect the phenolic compounds present in fruit peels. The almond kernels were blanched in boiling water, and the almond skin was oven-dried at 95-98oC for 10 h, as described by the authors. How did these heat treatments affect the soluble phenolic compounds?
- It appeared that the extract was used as a liquid in all experiments. How was the concentration determined because all analyses required an accurate concentration estimation? Were the TCP values used in the calculations? This needs to be specified clearly.
- What was the pH of the extract solution? As Table 3 shows lowered pH value for the test sample (almond extract) and little pH change as a function of time. If the amount of liquid extract was added at 1 g/1 kg meat (L188), it would need 80 mg/80 g of pork patties (Table 1), which would mean ~40 g of extract were needed for an 80 g patty. This would mean nearly 40 mL of extraction liquid were used for one patty!! While the positive control would contain 80 mg of sodium ascorbate powder. How could these two samples under such different packing conditions be compared?
- The results are not convincing in establishing a positive correlation between the antioxidant capacity of the almond extract and its use as a comparable antioxidant alternative for sodium ascorbate. There is no lipid-soluble phenolic compound in the almond extract. Why were the TBARS measurements used as the parameter? How would lipid oxidation be helpful as a measure?
- The results (section 3.3.4) showed no inhibitory activity in the almond extract against psychrotrophic microbial growth. There is no correlation between the antioxidative activities (Table 1) and antimicrobial properties. Why were these properties (Table 5) even studied?
- Writing: numerous grammatical mistakes and typos throughout the manuscript. For example:
- L40: weight, not "weigh"
- L41: discarded annually
- L42: filing, not "fill"
- L43: delete "in order"
- L44: delete "in order"
- L48: should be "lipids," and delete "to the basic chemical composition"
- L49: the total, not "total", and delete "existing", and delete "In this case, and change "different authors" to "several authors"
- L54-55: revise the whole sentence
- L57: delete "However"
- L58: change to "such as methanol", and change "hexane" to "and hexane"
- L59: delete "in order"
- L61: change "formulations" to "formulations," and delete "in use of", and change "green" to a "green"
- L62: change "represents" to "represent" and change "choise" to "choice"
- L63: delete "the best of" and "there is" and revise "the extraction of" to "extracting"
- L64: change "solvent" to "a solvent"
- L66: change "alternative" to "an alternative", and change "the recovery of" to "recovering"
- L72: change "specially" to "especially"
- L73: delete "the purpose of" and change "was" to "aimed", and change "as related" to "from"
- L74: delete "to be", and change "patties" to "patties,"
- L75: change "antioxidant" to "an antioxidant"
- L101: change "followed" to "follows"
- The whole "Conclusions" section needs to be rewritten as the results from this work do not support the claims.
Reviewer 2 Report
This study was to explore the function of almond skins obtained by green extraction and it's effect on the quality and shelf life of pork patties, which had some interesting points. But the data treatment and analysis, and the writing all needs to been further improved.
Abstract
1. line 20-22, “values of O2, a* and b* significantly decreased in pork patties”, in which treatment? All groups? Including control? Or just the treated groups?
Materials and methods
1. Line 182-183, was fat added in pork patties?
2. Line 188-192, why did you seal the package with a polyester methyl cellulose PLPMC film with oxygen permeability of 114 cm3/cm2/24 h, which was much higher than normal. So how can you know which caused the gas ratio changes during storage? Lipid oxidation? Microbiology activities? Or the permeation between outside? It’s difficult to explain. And please reconsider the analysis in line 384-395.
3. Line 193, was the samples also analyzed on day 0?
4. Line 246-247, how did you do the sample collecting of Salmonella spp. and L.monocytogenes?
5. Line 241-247, why did you incubate the Psychrotrophic microbial at 16 °C for 5 days? Many researches have used a much lower temperature (7 °C) to do this incubation. Please make sure.
Results and discussion
1. Line 405-414, how to explain the higher weight loss in CONTROL NEG, which had a higher pH?
2. Line 430-440, this part is not clearly elaborated. The correlation between L* and a* values was not necessary, in my opinion. And what’s the relationship between the lipid oxidation and L* value in this study?
3. Please check the significant level in a* values among different treatments on day 9. And why not had correlation analysis between L* and TBARS?
4. It was 10 gram in line 241, but it was 25 gram in line 488.
5. Change table 5 to figure to show the data. Change Log10 ufc/g to Log10 cfu/g in table 5.
Conclusion: needs further improvement.
Reviewer 3 Report
The idea of manuscript (Effect of phenolic compounds from almond skins obtained by green extraction on pork patties shelf life) is good and It tends to use a natural resource in the province pork patties shelf life. But I have some suggestions to improve the manuscript.
- In the introduction authors mentioned the almond contains phenols, flavonoids and anthocyanidins, So, why they only determine the phenols content, maybe the antioxidant activity of the extract may be due to flavonoids or anthocyanins. I suggest to determine the total flavonoids and total anthocyanins in the water extract, especially as author mentioned its first study in (green) water extract
- In material and methods line 97 (n=5) what it means is it 5 seeds or what
- Add the manufacturer, country and city of each device that was used
Results and discussion
- In Table 1, I suggest to revise the statistical analysis of ABTS results. I saw significance between the data numbers especially between the Antoñeta and other varieties. The author wrote (ns) no significance
- The discussion of Total phenol content must be increased
- In line 310-315, the authors wrote about significant correlations, where is the figure of these results. I suggest to add colored figures clear these analysis.
- In identification and quantification by HPLC-DAD analysis, authors found the highest quantities in all extracts while its glycosyloxyflavone compound and this confirms my request of the need to estimate the content of flavonoids and anthocyanins.
- I ask authors to add figure of phenolic compounds structure which identified by HPLC
- I ask authors to add the HPLC chromatogram to see the separation and peaks
- Authors mentioned in line 488 (Results showed the absence of pathogens (L. monocytogenes and Salmonella spp. in 25 g of 488 meat), Which results there are no any table or figure declare that
In conclusion:
- A sentence should be added to explain the practical importance of the research, the beneficiaries of it, and a future view of the work